# SAGE: Streaming Agreement-Driven Gradient Sketches for Representative Subset Selection

**Ashish Jha**
Skolkovo Institute of Science and Technology
`ashish.jha@skoltech.ru`

**Salman Ahmadi-Asl**
Research Center for Artificial Intelligence, Innopolis University
`s.ahmadiasl@innopolis.ru`

## Abstract

Training modern neural networks on large datasets is computationally and energy intensive. We present SAGE, a streaming data–subset selection method that maintains a compact Frequent Directions (FD) sketch of gradient geometry in $O(\ell D)$ memory and prioritizes examples whose sketched gradients align with a consensus direction. The approach eliminates $N \times N$ pairwise similarities and explicit $N \times \ell$ gradient stores, yielding a simple two-pass, GPU-friendly pipeline. Leveraging FD's deterministic approximation guarantees, we analyze how agreement scoring preserves gradient energy within the principal sketched subspace. Across multiple benchmarks, SAGE trains with small kept-rate budgets while retaining competitive accuracy relative to full-data training and recent subset-selection baselines, and reduces end-to-end compute and peak memory. Overall, SAGE offers a practical, constant-memory alternative that complements pruning and model compression for efficient training.

## 1 Introduction

The cost of training with all available data has grown sharply with the scale of modern benchmarks and models, motivating methods that reduce the number of examples processed without degrading generalization [1, 2, 3]. While pruning architectures, improving optimizers, and compressing models address complementary aspects of efficiency, the question of which examples most effectively drive learning remains central [4, 5]. Prior subset selection techniques either rely on heuristic uncertainty measures that ignore correlations between examples [6], or else approximate gradient matching using dense similarity computations that scale poorly with dataset size [7, 8, 9]. Recent advances such as gradient matching, submodular coverage, and influence- or value-based selection show that carefully chosen subsets can approach full-data performance [7, 9, 10, 11, 12], but they often require either $\Theta(N^2)$ pairwise computations or an explicit $N \times D$ gradient matrix limitations that hinder use at larger dataset scales [13, 7, 8].

We introduce SAGE, a method that sidesteps these scaling barriers by summarizing the evolving gradient rowspace with a deterministic Frequent Directions (FD) sketch [14, 15, 16]. The sketch provides a low-rank surrogate that preserves dominant gradient directions up to a quantified error, enabling one-pass, streaming operation with a memory footprint that scales only with the model dimension and the sketch size, not the number of training points [15, 16]. Within the sketched subspace, SAGE scores each example by the cosine agreement between its projected (normalized) gradient and the average projected direction. Selecting high-agreement examples yields subsets whose aggregate update tracks the full-data gradient while down-weighting inconsistent or noisy

39th Conference on Neural Information Processing Systems (NeurIPS 2025).

---

**Algorithm 1** SAGE: Streaming Agreement-Driven Subset Selection

---

1: **Input:** dataset $\mathcal{D}$, model $f_\theta$, loss $L$, sketch size $\ell$, target subset size $k$ (or per-class $k_c$)
2: Initialize $S \leftarrow 0_{\ell \times D}$, row counter $r \leftarrow 0$
3: **for** batches $B \subset \mathcal{D}$ **do**                          ▷ Phase I: streaming FD sketch
4:     **for** $(x, y) \in B$ **do**
5:         $g \leftarrow \nabla_\theta L(f_\theta(x), y); \quad S[r \bmod \ell, :] \leftarrow g; \quad r \leftarrow r + 1$
6:         **if** $r \bmod \ell = 0$ **then**
7:             $[U, \Sigma, V] \leftarrow \mathrm{svd}(S); \delta \leftarrow \sigma_\ell^2; \Sigma'_{jj} \leftarrow \sqrt{\max(\sigma_j^2 - \delta, 0)}$
8:             $S \leftarrow \Sigma' V^\top$
9:         **end if**
10:     **end for**
11: **end for**
12: *(Freeze $S$ for scoring)*
13: **Phase II:** For each example $i$, compute $z_i \leftarrow S g_i$; set $\hat{z}_i \leftarrow z_i / \|z_i\|_2$ if $\|z_i\|_2 > 0$ else $\hat{z}_i \leftarrow 0$
14: Compute $\bar{z} \leftarrow \frac{1}{N} \sum_i \hat{z}_i$; if $\|\bar{z}\|_2 > 0$, set $u \leftarrow \bar{z} / \|\bar{z}\|_2$, else $u \leftarrow 0$
15: Scores $\alpha_i \leftarrow \langle \hat{z}_i, u \rangle$
16: **if** class balance required **then**
17:     For each class $c$: $\bar{z}_c \leftarrow \frac{1}{|\mathcal{I}_c|} \sum_{i \in \mathcal{I}_c} \hat{z}_i; u_c \leftarrow \bar{z}_c / \|\bar{z}_c\|_2$ (if nonzero)
18:     Select top-$k_c$ per class by $\langle \hat{z}_i, u_c \rangle$   $(\sum_c k_c = k)$
19: **else**
20:     Select top-$k$ by $\alpha_i$
21: **end if**
22: **return** selected index set $T$

---

samples, echoing ideas from gradient-alignment literature [17]. This combination of streaming sketching and agreement-based ranking distinguishes SAGE from methods that rely on norms alone or expensive pairwise comparisons [18, 7, 8, 13]. We target regimes where per-example gradients are available or can be computed efficiently during training.

## 2   Method

Let $\{(x_i, y_i)\}_{i=1}^N$ denote the training set, $f_\theta$ a model with parameters $\theta \in \mathbb{R}^D$, and $\ell \ll N$ the sketch size. Denote by $g_i = \nabla_\theta L(f_\theta(x_i), y_i)$ the per-example gradient [19, 20]. SAGE maintains a Frequent-Directions (FD) sketch $S \in \mathbb{R}^{\ell \times D}$ updated in streaming fashion as gradients are observed [14, 15, 16]. When the sketch fills, compute a thin SVD $S = U\Sigma V^\top$, set $\delta = \sigma_\ell^2$, shrink $\Sigma'_{jj} = \sqrt{\max(\sigma_j^2 - \delta, 0)}$, and reconstruct $S \leftarrow \Sigma' V^\top$ [14, 15]. This contracts low-energy directions and retains dominant ones; letting $G \in \mathbb{R}^{N \times D}$ stack the $g_i^\top$ and $G_k$ be the best rank-$k$ approximation of $G$, FD guarantees

$$0 \preceq G^\top G - S^\top S \preceq \frac{2}{\ell} \|G - G_k\|_F^2 I \quad [15, 16].$$

**Two-pass scoring.**   After one streaming pass constructs $S$, SAGE performs a single additional backward pass to compute $z_i = S g_i \in \mathbb{R}^\ell$ [20]. For $z_i \neq 0$, define $\hat{z}_i = z_i / \|z_i\|_2$; let $\bar{z} = \frac{1}{N} \sum_{j=1}^N \hat{z}_j$ and (if $\|\bar{z}\|_2 > 0$) the unit consensus $u = \bar{z} / \|\bar{z}\|_2$. The agreement score is

$$\alpha_i = \langle \hat{z}_i, u \rangle \in [-1, 1],$$

which favors gradients aligned with the consensus direction and prevents high-magnitude outliers from dominating [17, 21]. SAGE selects the top-$k$ indices by $\alpha_i$, or a class-balanced variant (below). The entire procedure uses $O(\ell D)$ memory, independent of $N$ [14, 15].

### 2.1   Agreement Scoring and Selection

The projected gradients $z_i := S g_i$ define an agreement score $\alpha_i = \langle \hat{z}_i, u \rangle$ that measures each sample's contribution to the consensus direction of the sketched gradient distribution. Selection by top-$k$ $\alpha_i$ balances representativeness and diversity within the FD subspace; the class-balanced variant (*CB-SAGE*) replaces $u$ with per-class unit centroids $u_c$ and selects top-$k_c$ per class.

**Energy preservation.**

*Lemma* 1 (Consensus-direction energy). Let $T \subseteq [N]$ with $|T| = k$ and assume $\alpha_i \geq \xi > 0$ for all $i \in T$ and $\|\bar{z}\|_2 > 0$. Then

$$\sum_{i \in T} \langle z_i, u \rangle^2 = \sum_{i \in T} \|z_i\|_2^2 \, \alpha_i^2 \geq \xi^2 \sum_{i \in T} \|z_i\|_2^2.$$

*Proof.* Since $u$ is unit and $\alpha_i = \langle \hat{z}_i, u \rangle$, we have $\langle z_i, u \rangle = \|z_i\|_2 \, \alpha_i$; square and sum.

**Corollary (mean-alignment bound).** If a subset $T \subseteq [N]$ satisfies $\alpha_i \geq \xi > 0$ for all $i \in T$ and $\|\bar{z}\|_2 > 0$, then

$$\left\| \frac{1}{k} \sum_{i \in T} z_i \right\| \geq \xi \cdot \frac{1}{k} \sum_{i \in T} \|z_i\|_2.$$

*Proof.* Project the mean onto the unit consensus $u$: $\left\langle \frac{1}{k} \sum_{i \in T} z_i, u \right\rangle = \frac{1}{k} \sum_{i \in T} \|z_i\|_2 \alpha_i \geq \xi \cdot \frac{1}{k} \sum_{i \in T} \|z_i\|_2$, and $\|v\| \geq \langle v, u \rangle$ for unit $u$.

**Complexity.** (Excl. backprop) Two-pass epoch over $N$: $O(N \ell D + N \log k)$ time, $O(\ell D)$ memory; per-sample update & scoring are each $O(\ell D)$.

## 3 Experiments

We conduct experiments on five benchmark datasets spanning a range of computer vision tasks and data complexities - CIFAR-10, CIFAR-100, Fashion-MNIST, TinyImageNet, and Caltech-256. CIFAR-10 and CIFAR-100 are well-established datasets of $32 \times 32$ natural images with 10 and 100 classes respectively, providing a standard setting for controlled data selection analysis. Fashion-MNIST consists of grayscale images of clothing items across 10 categories, serving as a drop-in replacement for MNIST but with increased task difficulty. TinyImageNet is a subset of ImageNet with 200 classes and reduced image size, posing a more challenging scenario for both memory and accuracy. Caltech-256 contains 256 object categories with significant class imbalance and is used to demonstrate the robustness of subset selection methods in long-tailed data regimes.

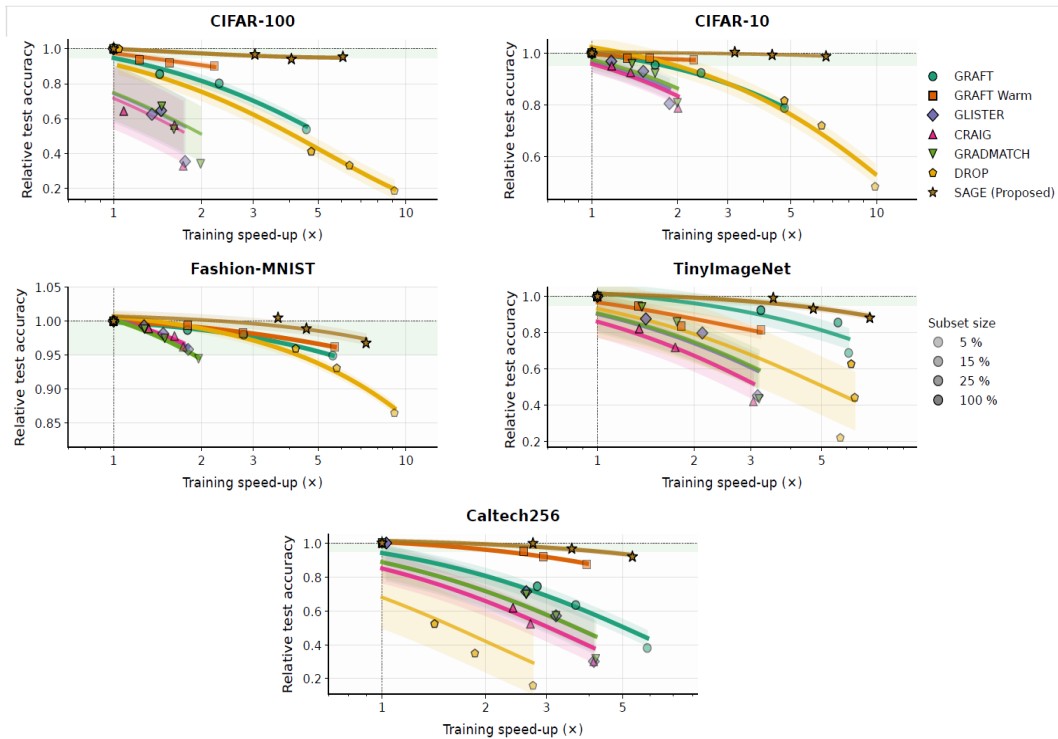

Figure 1: Figure 1: Relative test accuracy vs. training speed-up across CIFAR-10, CIFAR-100, Fashion-MNIST, TinyImageNet, and Caltech-256 at subset fractions of 5%, 15%, 25%, and 100%. SAGE achieves superior accuracy retention at aggressive subset fractions, matching or exceeding full-data accuracy at 25% data usage while delivering 3-6× training speed-ups. Curves show exponential fits with R² quality indicators, and shaded regions indicate variability across three independent seeds.

We compare SAGE against a comprehensive suite of baseline methods representative of the state of the art in data subset selection. These include GLISTER, which employs influence-based submodular optimization; CRAIG, which greedily maximizes gradient diversity using a submodular coverage function; GradMatch, a gradient matching-based selection method; and DROP, a scalable technique using a proxy for data importance. We also include GRAFT and GRAFT-Warm, which leverage gradient-aware fast maximum volume selection, to benchmark recent advances in the field. All methods are evaluated using open-source implementations and, where applicable, recommended hyperparameters. For each method–dataset pair, we evaluate performance by training a standard ResNet-18 using only the selected data subset. Subset fractions are set to $f \in \{0.05, 0.15, 0.25, 1.0\}$, corresponding to 5%, 15%, 25%, and 100% of the training set. We report top-1 test accuracy and wall-clock training time, measured on a single NVIDIA A100-80GB GPU. Each experiment is repeated three times with independent random seeds, and the mean and variance of performance are reported. All results are normalized relative to full-data training.

Our results demonstrate that SAGE consistently achieves higher accuracy retention at aggressive subset fractions and offers statistically significant acceleration in convergence compared to all baselines. On CIFAR-10 and TinyImageNet, SAGE matches or exceeds full-data accuracy using only 25% of the data, reducing training time by over $3\times$. On Caltech-256, which exhibits severe class imbalance, SAGE's class-balanced scoring improves subset representativeness and ensures uniform label coverage. Empirical response curves are modeled using a generalized exponential fit, and all results include $R^2$ fit quality and confidence bands for statistical rigor.

## 4 Related Work

Coreset selection and data pruning for efficient training have been explored via submodular coverage, gradient matching, and influence functions, among other ideas [12, 9, 7, 8, 10, 11]. CRAIG and

GLISTER exemplify selection by coverage and generalization proxies, respectively, while GRAD-MATCH formulates an explicit gradient-matching objective that scales quadratically in the number of examples [9, 8, 7]. More recent directions examine distributionally robust pruning and stepped greedy strategies aimed at favorable empirical trade-offs [22, 23, 24]. Complementary to these, GRAFT employs Fast MaxVol sampling on low-rank feature projections with dynamic gradient-alignment adjustments to perform in-training subset selection [25], while SAGE uses a Frequent Directions sketch with gradient-agreement scoring to perform streaming selection in constant memory and two passes, avoiding explicit $N^2$ similarities [25]. In parallel, Market-Based Data Subset Selection casts multi-criteria example utility into a convex cost/pricing framework, enabling principled aggregation of heterogeneous signals and tunable trade-offs [26]. However, many existing approaches either incur $\Theta(N^2)$ pairwise computations or require storing/operating on an explicit $N \times D$ gradient matrix, and several rely on bilevel or proxy objectives whose stability can vary across datasets.

SAGE differs by avoiding pairwise similarities [13, 7] and by using a deterministic sketch with worst-case guarantees [14, 15, 16], enabling a streaming implementation with memory constant in $N$ and only two sequential passes (sketch, then scoring) [15]. Within the sketched subspace, the agreement score operationalizes the intuition that *directional* consistency of gradients matters for optimization; unlike pure norm-based heuristics and early-prediction signals [18, 27], it explicitly aligns the subset's aggregate update with a consensus formed in the principal gradient subspace [17, 21]. A class-balanced variant further enforces label coverage without changing the constant-memory profile, making the method practical across both balanced and long-tailed regimes.

## 5  Limitations

SAGE assumes per-example (or microbatched) gradients and a second scoring pass, adding overhead vs. one-pass heuristics. FD incurs an additive $O(\|G - G_k\|_F^2/\ell)$ sketch error, so small $\ell$ can miss rare but important directions; agreement is directional (magnitude-agnostic) and may underweight hard examples.

## 6  Conclusion

We proposed SAGE, a streaming subset selection algorithm that couples Frequent Directions sketching with agreement-based ranking to identify representative training examples. The method offers theoretical control of sketch error, aligns selected gradients with a data-driven consensus, and scales to ImageNet-sized datasets on a single GPU. Experiments on multiple benchmarks show that SAGE preserves accuracy at aggressive pruning rates while delivering substantial speedups and memory savings, making it a practical tool for efficient model training.

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

# 7   Acknowledgement

The second author's work was supported by the Ministry of Economic Development of the Russian Federation (agreement No. 139- 10-2025-034 dd. 19.06.2025, IGK 000000C313925P4D0002).

# Supplementary Material

Table 1: Test accuracy (%) at subset fraction $f$. Best *non-full* entry in each column is bold; "Full data" is the 100% column.

| Method | CIFAR-100 | | | | TinyImageNet | | | |
|---|---|---|---|---|---|---|---|---|
| | 5% | 15% | 25% | 100% | 5% | 15% | 25% | 100% |
| Full data | – | – | – | 76.8 | – | – | – | 59.2 |
| Random | 45.1 | 59.3 | 65.7 | – | 28.4 | 42.1 | 49.5 | – |
| DROP | 48.9 | 63.2 | 68.1 | – | 31.7 | 46.8 | 52.3 | – |
| GLISTER | 52.1 | 66.7 | 70.5 | – | 35.2 | 49.1 | 54.6 | – |
| CRAIG | 53.8 | 67.9 | 71.8 | – | 36.8 | 50.5 | 55.9 | – |
| GradMatch | 55.3 | 69.1 | 72.6 | – | 38.4 | 51.7 | 56.8 | – |
| GRAFT | 56.9 | 70.2 | 73.5 | – | 39.6 | 52.9 | 57.4 | – |
| **SAGE** | **59.2** | **72.1** | **75.1** | – | **42.7** | **55.3** | **58.7** | – |

# Experimental Details

Backbone: ResNet-18 trained from scratch; optimizer: SGD+momentum 0.9, weight decay $5 \times 10^{-4}$, cosine LR; label smoothing 0.1, EMA 0.999; mixed precision enabled. Budgets $f \in \{0.05, 0.15, 0.25, 1.00\}$; selection frozen before training. Augmentations: random crop/flip (color jitter for TinyImageNet). Seeds: 3 per (dataset, budget, method); we report mean and 95% CI. Hardware: single NVIDIA A100-80GB; wall-clock measured end-to-end including selection. For class-imbalanced data we use CB-SAGE (per-class centroids). No code is released at submission; configs and exact commands will be provided post-review.

