# OpenReview forum: "SAGE: Streaming, Agreement-driven Gradient Sketches for Representative Subset Selection"
_NeurIPS.cc/2025/Workshop/Reliable_ML — NeurIPS 2025 - Reliable ML Workshop_

### Official Review · Reviewer_pUNi · 2025-09-13
**Needs more work in delivering the message, but the gem is hidden inside.**

**Rating:** 6
**Confidence:** 3

**Review:**

1. Summary. What the paper claims, how it does it, and the main results.

This paper suggests SAGE, an energy- and memory-efficient data-subset selection method to train neural networks on large datasets on small subset of data while keeping high accuracy. It computes a compact frequent directions sketch of gradient geometry, which is a low rank surrogate for gradient of each sample, instead of a N^2 pariwise computations. Using this information, SAGE computes cosine similarity between the projected gradient and the average direction, and only chooses high-agreement examples. The method was demonstrated on multiple computer vision datasets in comparion to other mini-batch subset selection methods, and it works well.

2. Strengths. Novelty, rigor, empirical/theoretical quality, clarity, relevance to reliability with imperfect data.
- the topic is timely
- Low-rank approximation makes sense

3. Weaknesses / Limitations. Missing comparisons/ablations, unclear assumptions, proof gaps, failure modes, scope limits.

- Not sure if this is about reliable ML
- The algorithm needs more explanation. Is your delta = sigma^2_l the largest or the smallest? (i am assuming smallest) Do you mean sigma^2_D, or do you mean l<D?
- Figure 1 is hard to comprehend. I cannot really distinguish the subset size (i think that's the darkness?) and I am not sure what it means by the training speed-up (x) in the x axis. Needs more elaboration about the figures.

4. Suggestions for Authors. Specific things that would improve the paper:

These are questions that would be nice if better clarified in the paper.

- The computational efficiency part should be more carefully addressed. If we already compute per-example gradients, what is the true computational advantage of selecting a subset? I think the advantage is to shorten the training rounds, although the computation of subset selection adds the cost. Although it may be obvious to the authors, it would be nice to articulate this in the paper.
- when do we get z_i = 0?

---

### Official Review · Reviewer_pSok · 2025-09-19
**Summary and feedbacks**

**Rating:** 7
**Confidence:** 2

**Review:**

This paper tries to solve an optimization problem for training neural networks. The authors propose a method called Streaming Agreement-Driven Gradient Sketches (SAGE) that uses streaming algorithms and gradient agreement to select a representative subset of the data for training. The method is evaluated on several image classification benchmarks, showing competitive accuracy and speed-ups compared to full-data training and other subset selection methods.
SAGE enables practical, GPU-friendly selection of representative subsets in a streaming fashion and achieves competitive accuracy and speed-ups on computer vision benchmarks while using a small fraction of the data.
SAGE approximates the evolving gradient geometry via a compact FD sketch, ranking examples by agreement with a consensus direction. This approach requires only two passes over the data, offers strict memory guarantees, and sidesteps pairwise computations or explicit gradient storage required by prior subset selection methods.
Through empirical evaluation on CIFAR-10/100, Fashion-MNIST, TinyImageNet, and Caltech-256, SAGE matches or exceeds full-data accuracy with just 25\% subset usage and achieves 3–6x training speed-ups compared to full-data training.
The author also gives a theoretical analysis that provides guarantees on energy preservation within the chosen subspace and empirical comparisons demonstrate SAGE’s advantages over state-of-the-art baselines (GLISTER, CRAIG, GradMatch, DROP, GRAFT).
For the weaknesses, the author assumes that the per-example gradients are accessible. which may not be practical in some scenarios. This work also relies on the FD sketch which might explode in some cases. The author should provide more analysis on the sketch size and its impact on the performance and how to select the sketch size.